# The PREDEP-SERT study protocol: A 6-month follow-up cohort study of predictors of effectiveness, tolerability and safety of sertraline for depression using Therapeutic Drug Monitoring

Patricio Molero[1,2]*, Covadonga Canga-Espina[1], Begoña Tapia-Alzuguren[3], José Pablo Bullard[1], María del Mar Unceta-González[1], Enrique Aubá[1,2], Jorge M. Núñez-Córdoba[4], Felipe Ortuño[1,2], Azucena Aldaz[3]*

1 Department of Psychiatry and Clinical Psychology, Clínica Universidad de Navarra, Pamplona, Navarra, Spain, 2 Instituto de Investigación Sanitaria de Navarra (IdiSNA), Pamplona, Navarra, Spain, 3 Pharmacy Services, Clínica Universidad de Navarra, Pamplona, Navarra, Spain, 4 Research Support Service, Central Clinical Trials Unit, Clínica Universidad de Navarra, Pamplona, Navarra, Spain

* pmolero@unav.es (PM); aaldaz@unav.es (AA)

## Abstract

Sertraline is a common first-line pharmacological treatment of Major Depressive Disorder (MDD). There is no established consensus nor clinical guidelines for personalized dose adjustments, which imply risks of toxicity and lack of efficacy. To address these challenges, there is preliminary evidence of the clinical utility of the determination of blood levels of sertraline by means of therapeutic drug monitoring (TDM). Further evidence is needed regarding the optimal therapeutic range of sertraline in terms of an optimal efficacy/tolerability-safety balance, with need of prospective studies on the association between blood concentration of sertraline and clinical outcomes. The PREDEP-SERT study (PREdictors of response in DEPression treated with SERTraline) is a single-center, observational, longitudinal, ambispective cohort study of patients with MDD to investigate the association between blood concentration of sertraline and its effectiveness, tolerability and safety (pre-registered study in Spain, REec number: 0014-2022-OBS, https://reec.aemps.es/reec/observacional/0014-2022-OBS). It adopts wide inclusion and exclusion criteria in order to allow the inclusion of patients that are representative of real-world clinical practice. It includes an exploratory retrospective subcohort and a subsequent 6-month follow-up prospective subcohort, with repeated measures (blood concentrations of sertraline and clinical outcomes) at scheduled timepoints (15 days, 30 days, 60 days, 90 days and 6 months). Relevant clinical, pharmacogenetic and sociodemographic potential predictors, confounders and effect modifiers will be explored. Its primary objective is to prospectively assess the association between blood concentration of sertraline (and the ratio with its metabolite, N-desmethylsertraline) and the intensity of depressive

**Data availability statement:** No datasets were generated or analysed during the current study. Data generated will not be shared publicly due to ethical and regulatory standards regarding confidentiality. Deidentified research data may be shared upon reasonable requests, upon study completion, via data use agreement as per the site's and local regulations.

**Funding:** The author(s) received no specific funding for this work.

**Competing interests:** PM reports (all outside the current work) having received research grants from the Ministry of Education (Spain), the Government of Navarra (Spain), the Spanish Foundation of Psychiatry and Mental Health and AstraZeneca; he has been a clinical consultant for MedAvanteProPhase and Worldwide Clinical Trials Limited and has received lecture honoraria from or has been a consultant for AB-Biotics, Adept Field Solutions, Dialectica, Guidepoint, Janssen, Novumed, Roland Berger, Scienta, and Survey Healthcare Global, received travel support for taking part in scientific meetings in the last 3 years (air/ground tickets + hotel) from Boston Scientific and Janssen, and has been the principal investigator of several studies supported by Janssen and Novartis about the efficacy and safety of novel pharmacological treatments for depression. This does not alter our adherence to PLOS ONE policies on sharing data and materials. The other authors declare no conflicts of interests.

symptoms measured by the Hamilton Depression Rating Scale at 6 months of treatment. Other secondary objectives are to assess the association between blood concentration of sertraline/N-desmethylsertraline and clinical variables of effectiveness (by means of validated clinical scales) and side effects at every timepoint. Its results may elucidate the clinical utility of TDM in the therapeutic management of MDD in a personalized fashion.

## 1. Introduction

Major depressive disorder (or depressive illness or clinical depression, thereafter referred to as depression) is a common mental disorder of public health concern with increased morbidity and mortality due to risk of suicide and other associated medical and psychiatric conditions, and is expected to be the leading cause of disease burden by 2030 [1,2]. Antidepressant medication is an integral part of the treatment for moderate to severe forms of depression, and selective serotonin reuptake inhibitors (SSRIs) are commonly recommended as first line options by clinical guidelines [3,4].

During the first weeks of treatment, a dose adjustment is made to identify the patient's minimum effective dose, which combines efficacy and acceptable tolerability. SSRI dose adjustment is made based on the established dose ranges in the summary of product characteristics and clinical judgment. Certain genetic and environmental factors can facilitate the development of toxicity or lack of efficacy within the established dose ranges, requiring the use of lower or higher doses. These situations imply the risk of, respectively, toxicity or excessively prolonged response latency, which can be associated with clinical complications. These situations are usually managed based on individual clinical judgment. There is no established consensus nor clinical guidelines for personalized dose adjustments. Therefore, determination of blood levels by means of therapeutic drug monitoring (TDM) may be determinant in this regard.

Among SSRIs and other antidepressant drugs, sertraline has been regarded as a strong candidate to be considered as the initial antidepressant for the treatment of major depression, considering the evidence for its efficacy, tolerability and acceptability [5]. This drug is also an example of an SSRI for which there is preliminary evidence of the importance of dose adjustment based on TDM, with optimal ranges of low or intermediate blood concentrations presumably associated with greater efficacy. Hence, increases beyond certain doses, even if within the established dose ranges in the summary of product characteristics, would not provide any clinical benefit. In a sample of 23 outpatients with recurrent depression followed during 1 year, a relationship was found between blood levels of sertraline and clinical improvement, with a trend to a curvilinear, inverted-U fashion, and lower blood levels of sertraline (25–50 ng/ml) being associated with adequate clinical maintenance treatment [6]. Absence of correlation between sertraline blood levels and global clinical improvement has been found in a sample of 21 outpatients at 30 days [7], though an U-shaped relationship was observed with specific symptom domains (blunted affect

and emotional withdrawal), specifically in the range of intermediate sertraline blood levels (40−70 ng/mL) [7]. By contrast, the established consensus of the therapeutic range of blood concentration of sertraline is very wide and includes higher concentrations, spanning from 10 to 150 ng/ml [8]. Oral dose of sertraline is associated with sertraline blood levels in a 1 year follow-up study (correlation coefficient $r = 0.37$ [6]), though this association was not found at 30 days [7]. Correlation coefficients of the relationship between sertraline blood levels and diverse clinical outcomes vary from −0.58 (with BPRS total scores at 30 days) [7], to 0.65 (with global clinical improvement at 1 year) [6]. In spite of an acceptable tolerability profile regarding its common adverse effects (AE), a recent pharmacovigilance analysis emphasizes the importance of caution regarding newly identified AEs, with focus in adolescents and elderly patients [9].

Further evidence is needed regarding the optimal therapeutic range of this drug in terms of the more favorable efficacy/tolerability-safety balance. There is a need for prospective studies focused on the association between blood concentration of not only sertraline, but also the ratio with its metabolite, N-desmethylsertraline, and depressive symptoms and clinical outcome in patients with depression treated with sertraline.

This study aims to assess whether there are optimum ranges of blood concentrations of sertraline associated with therapeutic response and acceptable tolerability and safety from the second week to the sixth month of treatment, and which relevant clinical, demographic and genetic factors may influence them. It is hypothesized that there is an association between blood concentration of sertraline and the intensity of depressive symptoms in the first 6 months of treatment, and that there are optimal ranges of blood concentration of sertraline to treat specific forms of depression (for instance, non-psychotic vs psychotic depression). Evidence of such ranges may facilitate the treatment of moderate to severe depression, demonstrate the clinical utility of TDM in its therapeutic management and serve as a case in point for other medications for mental illness.

## 2. Materials and methods

### 2.1. Study aim, design and setting

The PREDEP-SERT study (PREdictors of response in DEPression treated with SERTraline) aims to investigate the association between blood concentration of sertraline and its effectiveness, tolerability and safety in patients with depression under real-world setting. This will be a single-center observational, longitudinal, ambispective cohort study of patients (inpatients or outpatients of the site's psychiatry services) with major depressive disorder treated with sertraline that receive TDM per clinical judgment. The full cohort will comprise a retrospective subcohort and a prospective subcohort. The retrospective subcohort will include patients who already underwent TDM (without time limit) with registered values of blood concentration of sertraline in their clinical history, with the aim of an exploratory investigation to inform a subsequent prospective investigation. The prospective subcohort will have a 6-month follow-up period, with repeated measures (blood concentrations of sertraline and clinical outcomes) at scheduled timepoints (15 days, 30 days, 60 days, 90 days and 6 months). The recruitment center for both retrospective and prospective subcohorts will be an academic hospital.

### 2.2. Inclusion and exclusion criteria and sample size

This study adopts wide inclusion and exclusion criteria in order to allow the inclusion of patients that are representative of real-world clinical practice. The inclusion criteria are: 1) patients with a formal diagnosis of major depressive disorder, as main diagnosis or secondary to or comorbid with other medical conditions, confirmed by a psychiatrist, according to ICD-10 diagnostic criteria [10] (including the categories of moderate depressive disorder (F32.1), severe depressive episode with (F32.3) or without (F32.2) psychotic symptoms, recurrent depressive disorder, current episode moderate (F33.1), or severe, with (F33.3) or without (F33.2) psychotic symptoms). In the retrospective subcohort, this criterion may be met for patients with enough documented proof of consistent moderate or severe depressive symptoms, as main diagnosis or secondary to or comorbid with other medical conditions (as deemed by a psychiatrist); 2) who, per clinical judgment,

are being (prospective subcohort) or have been (retrospective subcohort) treated with sertraline and require (prospective subcohort) or have required (retrospective subcohort) TDM follow-up. The exclusion criteria are: 1) dysthymia as the sole diagnosis (as it may include a less severe profile of depressive symptoms); and 2) lack of adherence to the prescribed treatment with sertraline.

A minimum sample size of at least 61 patients in the prospective subcohort was calculated to achieve a statistical power of 80% to detect significant differences with an assumed correlation coefficient of 0,35 between blood concentration of sertraline and depressive symptoms (as a conservative estimation approach according to previous reports [6,7]), against a null hypothesis of absence of correlation, considering a two-tailed hypothesis test and a significance level of 5%. The adjusted sample size for an expected dropout rate of 10% will be 68 patients. This also will be the minimum sample size target for the retrospective cohort. Therefore, the anticipated sample size for the full cohort will be at least 136 patients.

## 2.3. Characteristics of participants and how the sample will be selected

Participants will be outpatients or inpatients, without age limit, with formal clinical diagnosis of depression treated with sertraline who receive TDM per clinical criterion. Potential prospective participants will be selected when a TDM is scheduled from the site's outpatient and inpatient psychiatry services where the attending psychiatrists involved in this study conduct their clinical activity (EA, CC, FO, JPB and PM). Retrospective data will be retrieved from consecutive prescriptions of sertraline TDM in the health records by means of an electronic, pseudonymized search conducted by personnel not involved in the study, following the site's Ethics Committee's procedure to access clinical records for research studies.

## 2.4. Description of all processes and interventions from informed consent and comparisons

Potential participants will be informed by their attending referring physicians if deemed appropriate for their best interest and overall well-being, in the context of their medical treatment with sertraline and follow-up with TDM. When a potential participant decides to participate in the prospective study, a clinical interview with one of the psychiatrists involved in this study will be conducted for informed consent and confirmation of eligibility criteria. The patient will receive the Informed Consent Form and will be informed that participation in the study is voluntary and that they may withdraw at any time without prejudice to their subsequent medical care. Throughout the study, all patients will continue the treatment with sertraline already indicated by their referring physicians, at the required dose per clinical judgment (official recommended doses in Spain: 50−200 mg per day [11]). Once the informed consent is signed, a baseline medical interview will be scheduled for formal diagnostic confirmation, assessment of psychiatric and general medical comorbidities, anthropometric, demographic and life style variables. Thereafter, medical follow-ups will be scheduled at 15 (+/-2) days, 30 (+/-5) days, 60 (+/-10) days, 90 (+/-15) days and 180 (+/-15) days including pharmacological variables and pharmacokinetic parameters of sertraline, effectiveness, tolerability and safety variables, variables of non-pharmacological treatments, analytic and genetic variables (if available per clinical judgement), suicide ideation or behavior, drugs use and other variables. Steady-state serum samples for the Therapeutic Drug Monitoring of sertraline will be obtained in fasting condition at first hour in the morning. After that, the patients will have the opportunity to have breakfast prior to the clinical assessment. Although only minimal and reasonable burden is added in the prospective subcohort relative to routine clinical practice to obtain relevant information, each clinical assessment may be broken into two sessions in the same day if the patient experiences fatigue. Patients' fatigue and cognitive capacity during the assessments will be taken into account by the psychiatrists conducting the assessment, and the opportunity to include a resting period at any timepoint of the assessment will be offered per clinical judgment, between two given clinical scales or parts of the psychiatric clinical interview. Upon completion of the study follow-up, participants will continue their medical treatment with their attending physicians of reference. Participants may withdraw from the study and continue their medical treatment exclusively with their attending physicians of reference at any time. Comparisons will be made between responders and non-responders to sertraline, and remitters

and non-remitters, at six months and every other timepoints. Response to sertraline at a given timepoint will be defined as a reduction of 50% or more of the total score in the depression rating scales (Hamilton Depression Rating Scale and Montgomery-Asberg Depression Rating Scale), and remission as a total score below the specific scale-thresholds.

**2.4.1. Therapeutic Drug Monitoring.** The protocol used for pharmacokinetic monitoring is established by the site's Pharmacy Service's Clinical Pharmacokinetics Unit (CPU) after a thorough review of the available scientific evidence and the relevant consensus published by the AGNP-TDM expert group [8]. Therapeutic Drug Monitoring (TDM) is generally conducted in either plasma or serum matrices, and there is currently no definitive consensus regarding the preferential use of one over the other. To date, definitive experimental evidence demonstrating significant differences in drug concentrations between plasma and serum is lacking, and the limited comparative data available suggest that both matrices may be used interchangeably for this purpose [8]. Our analytical determinations will be performed using serum samples specifically. Steady-state serum samples will be obtained from 3–5 mL of blood drawn, via venipuncture, into gel-free Vacutainer tubes. Sampling will be conducted at baseline time points (24 hours post-dose). Following centrifugation, serum will be processed the same day. If immediate processing is not feasible, samples will be stored at 2–8°C, protected from light, and analyzed within 24–72 hours. The quantification of serum concentrations of sertraline and its metabolite N-desmethylsertraline (N-SER) will be conducted by ultra-high performance liquid chromatography with tandem mass spectrometry (UPLC-MS/MS) at the CPU. To this end, the methodology outlined in the ChromeSystem commercial analysis kit (MassTox® TDM Series A Parameter Set Antidepressants 1/Extended) will be employed. This commercial kit comprises the following components: extraction and precipitation reagents, an internal standard, calibrators and quality controls, mobile phases, and a chromatographic column. The chromatographic equipment will be a Waters Acquity™ quadrupole which utilizes a positive electrospray ionization technique for the mass spectrometry. A calibration curve will be obtained prior to each sertraline and N-SER quantification.

**2.4.2. Specific genotyping techniques.** The pharmacogenetic analysis will be outsourced to the OneOme® LLC, a clinical laboratory located at 807 Broadway Street NE, Suite 100, Minneapolis, MN 55413. The analytical results will be obtained using tests that were developed and validated by OneOme, LLC, certified under CLIA-88 and accredited by the College of American Pathologists as qualified to perform high-complexity testing, approved for clinical use by the New York State Department of Health, with genomic DNA analyses by polymerase chain reaction (PCR) using Thermo Fisher TaqMan® and/or LGC Biosearch BHQ® probe-based methods [12].

Retrospective patients will be enrolled from the clinical health records and the values of sertraline concentrations (with the same protocol and specifications) and clinical data will be collected retrospectively using pseudonymized clinical data, which is exempt from informed consent as per the Spanish regulation RD 957/2020.

## 2.5. Variables and outcomes, clinical measures and timeline

This study will explore, in a cohort of patients with depression of a real word setting, the association between the exposure to blood concentrations of sertraline and clinical outcomes in terms of changes from baseline in the intensity of depressive symptoms (main outcome) and other secondary outcomes such as symptoms of anxiety and global clinical impression. Potential predictors, confounders and effect modifiers variables such as duration of current depressive episode, pharmacogenetic biomarkers, family history, sociodemographic characteristics, traumatic and/or stressful life events, drug use, symptoms of anxiety, obsessive-compulsive symptoms, suicide ideation and/or behavior, cognitive status and other non-pharmacologic concomitant treatments will be collected. Table 1 shows the planned schedule for assessments of the study.

The primary objective of this study is to prospectively assess the association between blood concentration of sertraline (and the ratio with its metabolite, N-desmethylsertraline) and the intensity of depressive symptoms measured by the Hamilton Depression Rating Scale (21-items) (HDRS) [13,14] at 6 months of treatment. Other secondary objectives are 1) to assess the association between blood concentration of sertraline and the variables of effectiveness: therapeutic

**Table 1. Schedule for assessments of the PREDEP-SERT study.**

|  | Variables | Timepoints | | | | | |
|---|---|---|---|---|---|---|---|
|  |  | Baseline | 15 (+/-2) days | 30 (+/-5) days | 60 (+/-10) days | 90 (+/-15) days | 180 (+/-15) days |
| Psychiatric[a] and other medical diagnoses | Main diagnosis, psychiatric comorbidities, other medical comorbidities | x |  |  |  |  |  |
| Pharmacokinetic variables (sertraline TDM[b]) | Date and time of extraction, serum concentrations of sertraline and n-desmethylsertraline | x[c] | x | x | x | x | x |
| Medication adherence | Triple confirmation[d] | x | x | x | x | x | x |
| Effectiveness | HDRS[e], MADRS[f], HARS[g], GCI[h], BDI[i], C-SSRS[j], Y-BOCS[k], MMSE[l] | x | x | x | x | x | x |
| Anthropometric | age, weight, height, body mass index, body surface | x | x | x | x | x | x |
| Demographic and lifestyle | sex, marital/civil status, place in the phratry, education, caregiver (siblings, dependent/sick others), religious beliefs/community religious activity (yes/no), social activity (maintained/impoverished/none-loneliness), residence (urban/rural) | x |  |  |  |  |  |
| Drug use | Tobacco, alcohol, cannabis, cocaine, amphetamines, others | x | x | x | x | x | x |
| Physical activity (aerobic, anaerobic) | Sedentarism (<30 min/week), 30–150 min/week, 150–300 min/week, >300 min/week. | x | x | x | x | x | x |
| Diet | Healthy, Unhealthy[m] | x | x | x | x | x | x |
| First degree psychiatric family history | Depression, bipolar disorder, psychotic disorders, others. | x |  |  |  |  |  |
| CYP[n] isophorms (CYP 2B6, CYP 2C19, CYP 2D6) | PM, IM, NM, RM, UM[o] | x |  |  |  |  |  |
| Work activity | Student, unemployed, employed, self-employed, disability, retired[p] | x |  |  |  |  |  |
| Night shifts (% of work activity) | 0%, 25%, 50%, 75%, 100% | x |  |  |  |  |  |
| Acute, current stressful life events | Frequency | x |  |  |  |  |  |
| Chronic, current stressful life events | Frequency | x |  |  |  |  |  |
| Current, traumatic life events | Frequency | x |  |  |  |  |  |
| Past, traumatic life events (lifetime) | Frequency | x |  |  |  |  |  |
|  |  | x |  |  |  |  |  |
| Expectations[q] | Worsen, none/neutral, improvement | x |  |  |  |  |  |
| Insight[r] | No insight, partial insight, insight | x |  |  |  |  |  |
| Past active suicide ideation[s] | No, yes | x |  |  |  |  |  |
| Past suicide behavior[t] | No, yes | x |  |  |  |  |  |
| Past non-suicidal self harm[u] | No, yes | x |  |  |  |  |  |
| Duration of the current depressive episode | Months | x |  |  |  |  |  |
| Time from initial diagnosis to the current episode | Years | x |  |  |  |  |  |

*(Continued)*

| | Variables | Timepoints | | | | | |
|---|---|---|---|---|---|---|---|
| | | Baseline | 15 (+/-2) days | 30 (+/-5) days | 60 (+/-10) days | 90 (+/-15) days | 180 (+/-15) days |
| Number of previous depressive episodes | | x | | | | | |
| Adverse effects | UKU[v] | x | x | x | x | x | x |
| Quality of life | QLDS[w] | x | x | x | x | x | x |
| Non-pharmacologic treatment modalities | Promotion of healthy lifestyle, psychotherapy, electro-convulsive therapy or other neurostimulation therapies | x | x | x | x | x | x |
| Analytical variables[x] | | x | x | x | x | x | x |

[a] According to ICD-10 diagnostic criteria.

[b] Therapeutic Drug Monitoring of Sertraline and N-desmethylsertraline concentrations in blood (seric concentrations).

[c] Only if available at baseline (patients who started sertraline and had a TDM scheduled on the baseline visit.

[d] Triple confirmation of adherence: verification of active medical prescription, self-reported confirmation by the patient (by means of the Medication Adherence Rating Scale (MARS) and confirmation by caregiver or significant other (whenever it is possible and with prior patient's authorization).

[e] Hamilton Depression Rating Scale (21-items) (HDRS).

[f] Montgomery-Asberg Depression Rating Scale (MADRS).

[g] Hamilton Anxiety Rating Scale (HARS).

[h] Global Clinical Impression (GCI).

[i] Beck Depression Inventory (BDI).

[j] Columbia-Suicide Severity Rating Scale (C-SSRS).

[k] Yale-Brown Obsessive-Compulsive Scale (Y-BOCS).

[l] Mini-mental State Examination (MMSE), at baseline and at 6 months.

[m] Healthy diet: varied and balanced dietary pattern, of which a significant proportion of energy intake comes from fruits, vegetables, nuts, and whole grains, while the intake of (saturated) fats and free sugars is relatively low. Unhealthy diet: dietary pattern with an opposite composition [32].

[n] Cytochromes P450. If pharmacogenetic analysis are available per clinical judgment.

[o] PM: poor metabolizer; IM: intermediate metabolizers; NM: normal metabolizer; RM: rapid metabolizer; UM: ultrarapid metabolizer.

[p] As collected in the clinical records.

[q] Expectations of patients about their pharmacological treatment.

[r] Awareness of experiencing a mental disorder.

[s] Any thoughts, of any intensity or duration, of wanting to actively end one's life by suicide, with or without methods or intent (excluding passive suicide ideation such us wish to be death, or not alive anymore or not wake up).

[t] A potentially self-injurious act undertaken with at least some wish to die, as a result of act (without need of actual injury or harm).

[u] Any actual injury or harm (of any severity, including very minor such as surface scratches) without any (=0) intent to die as a result of it.

[v] Type, severity and attribution of adverse effects, by means of the UKU side effects rating scale.

[w] By means of the Quality of Life in Depression Scale (QLDS).

[x] If available per clinical judgment: Complete blood count, ionogram, vitamin B1, B6, B12, folate, ferritin, plasma albumin, ALT, AST, total and direct bilirubin, alkaline phosphatase, urea and creatinine.

response (reduction of 50% of the intensity of depressive symptoms from baseline measured by the HDRS) and remission (HDRS score ≤ 7 points) at 6 months of treatment; global symptomatic profile at different timepoints (15 days, 30 days, 60 days, 90 days and 6 months) measured by the HDRS, the Montgomery-Asberg Depression Rating Scale (MADRS) [15,16] (less influenced by anxiety or physical symptoms than the HDRS, for which therapeutic response is also a reduction of 50% of the total score from baseline, and remission is a total score ≤ 6 points), Beck Depression Inventory (BDI) [17,18] (which evaluates cognitive symptoms of depression), the Hamilton Anxiety Rating Scale (HARS) [16,19], Global

Clinical Impression (GCI) [20], suicide ideation and/or behavior by means of the Columbia-Suicide Severity Rating Scale (C-SSRS) [21,22], obsessive-compulsive symptoms by means of the Yale-Brown Obsessive-Compulsive Scale (Y-BOCS) [23,24], cognitive assessment at baseline and at 6 months of treatment (Mini-mental State Examination (MMSE) [25,26] or additional tests if conducted); and adverse effects (type, severity and attribution, by means of the UKU side effects rating scale [27]); 2) describe the utility of the combination of pharmacogenetic biomarkers relevant for sertraline (if pharmaco-genetic analysis are available per clinical judgment, including polymorphisms of CYP2B6, 2D6 and 2C19 isoforms) and pharmacokinetic monitoring for the optimization of the pharmacological treatment of depression; 3) describe the influence on the clinical outcome an blood concentrations of setraline of psychiatric family history (in first degree relatives), drug use, lifestyle (diet, physical, social and religious activity), and work, social and family conditions (at baseline, related variables described in Table 1); 4) determine the implication in the clinical outcome an blood concentrations of setraline of non-pharmacologic treatment modalities (promotion of healthy lifestyle, psychotherapy, electroconvulsive therapy or other neurostimulation therapies), in addition to the intake of co-medications, complementary and alternative treatments including herbal remedies, and specific medically prescribed diets (such as low-protein diet) (baseline and all timepoints, related variables described in Table 1).

Other variables will include expectations of patients about their pharmacological treatment (of worsening, none/neutral, improvement); insight (awareness of experiencing a mental disorder); medication adherence: given its importance as the sole exclusion criteria, adherence to the prescribed medication will be assessed by triple confirmation: verification of active medical prescription, self-reported confirmation by the patient (by means of the Medication Adherence Rating Scale (MARS) [28] and confirmation by caregiver or significant other (whenever it is possible and with prior patient's authorization); quality of life: will be measured in every visit by means of the Quality of Life in Depression Scale (QLDS) [29,30]; duration of the current depressive episode; time from initial diagnosis to the current episode; number of previous depressive episodes; analytical variables (at baseline and all timepoints, if available per clinical judgment); work activity and night shifts; traumatic and/or stressful life events (acute and/or chronic). All the clinical scales will be used in their validated versions in Spanish collected in a reference bank of basic instruments for the practice of clinical psychiatry [31]. Of note, all the clinical and demographic variables to be assessed in the prospective subcohort have been selected taking into consideration the routine clinical practice of an academic hospital which includes the core variables of this study: the routine use of TDM and pharmacogenetic analyses per clinical judgment, the specified psychometric scales (HDRS, MADRS, HARS, C-SSRS, Y-BOCS and MMSE), and thorough clinical interviews and documentation that allow to retrieve the same core information in the retrospective subcohort. Only minimal and reasonable burden is added in the prospective subcohort relative to routine clinical practice, to obtain relevant information such as the CGI, the BDI, the UKU side effects rating scale, the MARS, the QLDS and the exploration of patients' expectations about their pharmacological treatment. These latter variables are expected to be available only in the prospective subcohort.

## 2.6. Data management plans, type of data and statistical analyses planned

The data source for all the demographic and clinical variables will be the psychiatric health records. The blood samples for the determination of blood concentration of sertraline and its metabolite, N-desmethylsertraline, will be obtained from new patient's lab extractions per clinical practice, and their final destination will be destruction. Retrospective data will be obtained by means of pseudonymization following the site's ethically approved procedure. To ensure data confidentiality, patients in the study will be identified by a sequential with a number, consecutively according to the order of inclusion. Data collection forms, reports, and study communications will be identified with this number. Confidentiality of participants' data will be guaranteed in compliance with ethical and regulatory standards.

The retrospective and prospective data will be analyzed separately. The retrospective cohort is expected to be completed and analyzed prior than the prospective cohort. Both subcohorts will be also combined for analysis of the full cohort. The statistical analysis plan will include a descriptive analysis for summarizing study variables, including the

calculation of means and standard deviations for quantitative variables and percentages for qualitative variables. Pearson's correlation coefficient will be used to assess the association between plasma sertraline concentration and the intensity of depressive symptoms. The odds ratio and its 95% confidence interval will be calculated as a measure of association using logistic regression models. These models will be used to assess specifically the associations between blood concentration of sertraline (independent variable) and variables of effectiveness (therapeutic response and remission according to HDRS and MADRS), suicide ideation and/or behavior by means of the C-SSRS, and potential side effects (presence or absence). P values <0.05 will be considered to establish statistical significance. Statistical analyses (including interactions and sensitivity analyses) and will be performed using the available Stata versions in our site, currently up to Stata 19 (StataCorp. 2025. Stata Statistical Software: Release 19. College Station, TX: StataCorp LLC). No imputation for missing data will be applied. To maximize the data collection, quality control reviews will be conducted after every visit. A flexible time-window for visits (Table 1) and periodic reminder phone calls will be planned to reduce the risk of follow-up losses.

### 2.7. Safety considerations

This study includes a plan for the management and reporting of adverse reactions and other relevant events. Suspected cases of serious and unexpected adverse reactions to sertraline will be reported through the official channel provided to site's computerized system and medical records. Tolerability and safety related information relevant to the study will be collected. This study does not modify the therapeutic regimen of the participants, which includes the obligations inherent to the administration of medications and the detection and reporting of these to pharmacovigilance systems when appropriate.

### 2.8. Ethical considerations and declarations

This protocol has been reviewed and approved by the local Research Ethics Committee, which complies with the international standards of GCP CPMP/ICH/135/95 (Comunidad Foral de Navarra Research Ethics Committee; reference code: PREDEP-SERT; EO_2021/19) and will be conducted in accordance with the Declaration of Helsinki. This study is registered in the Spanish Registry of Clinical Studies (Registro Español de Estudios Clínicos (REec) Identifier: 0014-2022-OBS) (https://reec.aemps.es/reec/observacional/0014-2022-OBS). All prospective participants will need to sign the approved informed consent form in order to be enrolled. The informed consent includes the collection and use of the data and biological specimens of the participants, permission to publish their clinical data, and future publications of clinical data of the participants does not compromise anonymity or confidentiality or breach local data protection laws. The informed consent for minors includes consent from their parents, and minors aged 12 and above should also provide their assent as per national regulation. Therefore, this study includes three different versions of the document including the subject information sheet and informed consent form (all approved by the local Research Ethics Committee): one for adults, a different one for minors, and an additional one for minors aged 12 or above to document assent in that age group. As for the retrospective subcohort, this study uses pseudonymized clinical data, authors cannot identify individual participants during or after data collection and is exempt from informed consent as per the Spanish regulation RD 957/2020. The inclusion criteria include a main diagnosis for which sertraline is a pharmacological treatment of first choice, and a follow-up period (6 months) shorter than the usual minimum duration of treatment has been stablished. This study protocol follows the applicable items of the STROBE Checklist [33].

### 2.9. Status and timeline of the study

This study was approved on 25/08/2021, and started on 29/12/2021. Recruitment and data collection, especially in the prospective cohort, were hampered by the consequences of the COVID-19 pandemic. On 27/03/2024 a protocol amendment was approved (protocol version 3 of 06/March/2024, as supplementary material) to increase representativity by

excluding age limits and including depressive disorders secondary to medical or psychiatric comorbidities. Recruitment and data collection are ongoing and expected to continue until December 2026. Analysis and interpretation of the retrospective data obtained are ongoing and initial retrospective results are expected to be sent for publication by mid-end-2025. End of follow-up of all prospective subjects is expected for June 2027. Analysis, interpretation and results of the prospective cohort are expected for September 2027, once the required number of participants of the prospective cohort have been recruited and have completed the 6-month follow-up period.

## 3. Discussion

### 3.1. Limitations of the study design

This study has the limitations inherent to an observational study design. One potential major limitation is the risk of bias due to confounding variables. This study will gather enough information to control for the most relevant confounding factors in the statistical analysis phase of the study.

Furthermore, the adherence of the retrospective subcohort to planned timepoints may depend on the availability of data. This may result in lack of homogeneity between retrospective and prospective subcohorts. To address this limitation, separate analysis for both subcohorts will be also conducted. In addition, the timepoints of the retrospective cohort will be adjusted as much as possible to the flexible time-window planned for the prospective cohort. Of note, the measure of the severity of depression in both the retrospective and prospective subcohorts will be the HDRS, given that this scale is routinely administered and documented in psychiatric health records per clinical practice in our site. With this approach we expect to increase the homogeneity between the retrospective and prospective subcohorts for the primary objective of this study.

Another limitation is related with the difficulty of clinical scales to capture the full spectrum of the clinical phenomenology of the depressive illness. For that reason, a formal diagnosis of clinical depression according to the ICD diagnostic criteria confirmed by a psychiatrist will be required in the inclusion criteria, opting mainly for clinician-administered scales (as opposed to self-rated scales). Three specific scales of clinical depression have been included to capture all the clinical domains as much as possible, and also to overcome the fact that the HDRS has been deemed too much influenced by symptoms of anxiety and physical symptoms, and therefore with limitations in patients with comorbid medical conditions [31], as is expected to be the case in this cohort. Finally, no validated clinical scale for evaluating medication adherence specifically in depression is available to our knowledge. The Medication Adherence Rating Scale (MARS) will be used as a reasonable proxy for this purpose, although this tool was originally designed for medication adherence in psychosis.

### 3.2. Dissemination plans

The results will be submitted for publication to peer-reviewed scientific journals in a two-step fashion: the results from the retrospective cohort will be sent for publication in a first instance. Once the follow-up of the prospective cohort is completed, its results (expected for September 2027) will be analyzed and published in a second step, together with an interpretation of the overall results. The original (and English translation) of the study are openly available as supplementary materials.

### 3.3. Managements of amendments to the study

Any substantial modification to the study protocol will be properly justified and communicated to relevant parties to comply with ethical and regulatory requirements.

## Supporting information

**S1 File. Original protocol.**
(DOC)

**S2 File.  English translation of the original protocol.**
(DOCX)

**S3 File.  STROBE checklist of cohort studies.**
(DOC)

## Acknowledgments

We gratefully acknowledge the institutional support for this research project.

## Author contributions

**Conceptualization:** Patricio Molero, Azucena Aldaz.

**Methodology:** Patricio Molero, Covadonga Canga-Espina, Begoña Tapia-Alzuguren, José Pablo Bullard, María del Mar Unceta-González, Enrique Aubá, Jorge M. Núñez-Córdoba, Felipe Ortuño, Azucena Aldaz.

**Project administration:** Patricio Molero, Azucena Aldaz.

**Resources:** Azucena Aldaz.

**Supervision:** Patricio Molero, Azucena Aldaz.

**Writing – original draft:** Patricio Molero.

**Writing – review & editing:** Patricio Molero, Covadonga Canga-Espina, Begoña Tapia-Alzuguren, José Pablo Bullard, María del Mar Unceta-González, Enrique Aubá, Jorge M. Núñez-Córdoba, Felipe Ortuño, Azucena Aldaz.

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
