## [Decision Letter · Decision Letter 0]

26 Jun 2025

Dear Dr.  Molero,

Thank you for submitting your manuscript to PLOS ONE. After careful consideration, we feel that it has merit but does not fully meet PLOS ONE’s publication criteria as it currently stands. Therefore, we invite you to submit a revised version of the manuscript that addresses the points raised during the review process.

We look forward to receiving your revised manuscript.

Kind regards,

Dickens Akena, Ph.D

Academic Editor

PLOS ONE

Journal Requirements:

2. Thank you for stating the following in the Competing Interests section: [PM reports (all outside the current work) having received research grants from the Ministry of Education (Spain), the Government of Navarra (Spain), the Spanish Foundation of Psychiatry and Mental Health and AstraZeneca; he is a clinical consultant for MedAvanteProPhase and Worldwide Clinical Trials Limited and has received lecture honoraria from or has been a consultant for AB-Biotics, Adept Field Solutions, Dialectica, Guidepoint, Janssen, Novumed, Roland Berger, and Scienta, received travel support for taking part in scientific meetings in the last 3 years  (air/ground tickets + hotel) from Boston Scientific and Janssen, and has been the principal investigator of several studies supported by Janssen and Novartis about the efficacy and safety of novel pharmacological treatments for depression. The other authors declare no conflicts of interests.].

4.Please review your reference list to ensure that it is complete and correct. If you have cited papers that have been retracted, please include the rationale for doing so in the manuscript text, or remove these references and replace them with relevant current references. Any changes to the reference list should be mentioned in the rebuttal letter that accompanies your revised manuscript. If you need to cite a retracted article, indicate the article’s retracted status in the References list and also include a citation and full reference for the retraction notice.

Reviewers' comments:

Reviewer's Responses to Questions

**Comments to the Author**

1. Does the manuscript provide a valid rationale for the proposed study, with clearly identified and justified research questions?

Reviewer #1: Yes

Reviewer #2: Yes

2. Is the protocol technically sound and planned in a manner that will lead to a meaningful outcome and allow testing the stated hypotheses?

Reviewer #1: Partly

Reviewer #2: Yes

3. Is the methodology feasible and described in sufficient detail to allow the work to be replicable?

Reviewer #1: Yes

Reviewer #2: Yes

4. Have the authors described where all data underlying the findings will be made available when the study is complete?

Reviewer #1: Yes

Reviewer #2: Yes

5. Is the manuscript presented in an intelligible fashion and written in standard English?

Reviewer #1: Yes

Reviewer #2: Yes

You may also provide optional suggestions and comments to authors that they might find helpful in planning their study.

Reviewer #1: Reviewer’s report

Study Title: The PREDEP-SERT study protocol: a 6-month follow-up cohort study of predictors of effectiveness and tolerability of sertraline for depression using Therapeutic Drug Monitoring

This study aims to prospectively assess the association between blood concentrations of sertraline and its metabolite with the severity of depressive symptoms after six months of treatment. Additionally, it seeks to examine the relationship between sertraline concentrations and key clinical outcomes of effectiveness and side effects. This is an important study with the potential to inform optimal sertraline dosing in real-world clinical practice. The inclusion of a study population representative of routine clinical settings enhances the relevance and applicability of the findings. Below are a few suggestions to further strengthen the manuscript.

Comment 1: The primary objective of the study is to examine the association between blood levels of sertraline (and its metabolite) and the severity of depressive symptoms. For the retrospective cohort, it would be helpful to clarify how the severity of depression will be measured using existing medical records. Are there depression assessment tools that are routinely administered and documented in patient charts, and which you expect to retrieve for this purpose?

Comment 2: The variables to be assessed in the prospective cohort are clearly outlined. However, it would be helpful to specify which variables are available (or expected to be available) in the retrospective cohort.

Comment 3: The analysis plan is rather very brief and would benefit from further elaboration. Specifically:

- What statistical analyses are planned for the retrospective cohort, given the nature and possible limitations of the data available in the medical records?

- More detail on the logistic regression models would be helpful. Which specific associations are you aiming to explore, and what variables will be included in the models?

- One of the secondary objectives is to assess the association between blood sertraline concentrations and other clinical variables related to effectiveness and side effects. What is the analysis plan for this objective?

Reviewer #2: Thank you for the opportunity to review this important and timely research study. The study presents a comprehensive exploration of bio-psycho social model in the management of patients with depression and offers valuable insight into personalized sertraline treatment approaches, particularly through genetic profiling. This is a relevant and forward-looking topic that could meaningfully contribute to both clinical practice and future research.

Minor Comments

1. Terminology Clarification – "Tolerability" vs. "Drug Safety"

In both the abstract (line 36) and the methods section (line 120), the authors appear to use "tolerability" and "drug safety" interchangeably. However, these terms have distinct meanings in pharmacological research and should be clearly differentiated.

• Drug safety refers to the broader evaluation of all potential adverse effects, including serious adverse events and long-term risks, and entails a comprehensive risk–benefit analysis.

• Drug tolerability, on the other hand, centers on the subjective experience of the patient, especially non-serious side effects that affect adherence or comfort.

I recommend that the authors revise these sections to reflect the appropriate term depending on the specific outcome they are measuring, or explicitly state whether both aspects are being evaluated.

2. Feasibility of Assessments Across Age Spectrum

The methods describe multiple assessment tools administered in a single session. Given that the study includes participants across a broad age range including older children and the elderly. it would be helpful to clarify how the burden of lengthy assessments (potentially exceeding two hours) will be managed. Specifically:

• Will assessments be broken into phases or sessions for specific age groups?

• How will participant fatigue or cognitive capacity in extreme age groups be accounted for?

Clarifying the administration logistics would strengthen the feasibility of the methodology.

3. Ethical Considerations – Informed Consent and Assent

The manuscript states that informed consent will be obtained from adult participants and the parents/guardians of minors. However, it is important to also address assent, particularly for participants aged 12 and older.

Under Spanish regulations (Law 41/2002), while parental consent is required for minors, children aged 12 and above should also provide their assent. The authors should clarify whether and how assent will be obtained in accordance with these ethical requirements.

4. Data Sharing Policy

The data management plan is well written, but the data sharing statement could be improved. While the authors indicate that data will be available upon request, it is advisable to align with current data sharing policies, which encourage more transparent and accessible data availability. A clear statement on how and when data will be shared (e.g., via a repository or data use agreement) would enhance reproducibility and adherence to journal policies.

seamless Comments

• The discussion is clearly written and thoughtfully addresses the limitations of the study.

• The distinction between retrospective and prospective cohorts is appropriately acknowledged, and the plan to analyze them separately is sound given the potential for missing variables in retrospective data.

Conclusion

This is a well-conceived study that contributes meaningfully to the literature on depression management and personalized medicine. With clarification on terminology, ethical procedures, and data sharing, the manuscript will be greatly strengthened. I commend the authors for undertaking this valuable work.

**Do you want your identity to be public for this peer review?** For information about this choice, including consent withdrawal, please see our Privacy Policy

Reviewer #1: No

Reviewer #2: **Yes: ** Dr. Namuli Justine Diana

---

## [Author Response · Author response to Decision Letter 1]

27 Jun 2025

Pamplona, Spain, June 27, 2025

Dr. Dickens Akena

Dear Dr. Akena,

We truly thank the effort, time and dedication devoted by the Reviewers and yourself as the Academic Editor to this manuscript. All the insights, comments and suggestions proposed have been carefully considered and well taken into account, and have certainly contributed to improve this article, which has been amended accordingly.

In this letter we outline every change made in response to the Journal Requirements and the Reviewers comments.

Yours sincerely, also on behalf of our co-authors, Patricio Molero and Azucena Aldaz

Patricio Molero M.D., Ph.D. Department of Psychiatry and Clinical Psychology. Clínica Universidad de Navarra, Av. De Pio XII 36, 31008, Pamplona, Navarra, Spain. +34948255400; pmolero@unav.es

Azucena Aldaz, Pharm.D., Ph.D. Pharmacy Services. Clínica Universidad de Navarra, Av. De Pio XII 36, 31008, Pamplona, Navarra, Spain. +34948255400; aaldaz@unav.es

Journal Requirements:

1. We have ensured that our manuscript meets PLOS ONE's style requirements, including those for file naming.

2. We confirm that the stated Competing Interests do not alter our adherence to all PLOS ONE policies on sharing data and materials. Therefore, the updated Competing Interests statement should read like this, please:

“PM reports (all outside the current work) having received research grants from the Ministry of Education (Spain), the Government of Navarra (Spain), the Spanish Foundation of Psychiatry and Mental Health and AstraZeneca; he has been a clinical consultant for MedAvanteProPhase and Worldwide Clinical Trials Limited and has received lecture honoraria from or has been a consultant for AB-Biotics, Adept Field Solutions, Dialectica, Guidepoint, Janssen, Novumed, Roland Berger, Scienta, and Survey Healthcare Global, received travel support for taking part in scientific meetings in the last 3 years (air/ground tickets + hotel) from Boston Scientific and Janssen, and has been the principal investigator of several studies supported by Janssen and Novartis about the efficacy and safety of novel pharmacological treatments for depression. This does not alter our adherence to PLOS ONE policies on sharing data and materials. The other authors declare no conflicts of interests.”

3. Please note that we adhere to your open data policy. The entire data of this Protocol will be made freely accessible if our manuscript is accepted for publication. This means that the data associated with this work (namely: the full Original protocol approved by our local research ethics board, in Spanish, and its translation to English) have been attached as Supporting Information to be freely accessible upon publication.

4. The reference list has been reviewed to ensure that it is complete and correct.

Response to reviewers

Reviewer #1: Reviewer’s report

Study Title: The PREDEP-SERT study protocol: a 6-month follow-up cohort study of predictors of effectiveness and tolerability of sertraline for depression using Therapeutic Drug Monitoring

This study aims to prospectively assess the association between blood concentrations of sertraline and its metabolite with the severity of depressive symptoms after six months of treatment. Additionally, it seeks to examine the relationship between sertraline concentrations and key clinical outcomes of effectiveness and side effects. This is an important study with the potential to inform optimal sertraline dosing in real-world clinical practice. The inclusion of a study population representative of routine clinical settings enhances the relevance and applicability of the findings. Below are a few suggestions to further strengthen the manuscript.

Response: We thank Reviewer #1 very much.

Comment 1: The primary objective of the study is to examine the association between blood levels of sertraline (and its metabolite) and the severity of depressive symptoms. For the retrospective cohort, it would be helpful to clarify how the severity of depression will be measured using existing medical records. Are there depression assessment tools that are routinely administered and documented in patient charts, and which you expect to retrieve for this purpose?

Response: We thank Reviewer #1 very much for this observation. Of note, the measure of the severity of depression in both the retrospective and prospective subcohorts will be the Hamilton Depression Rating Scale (HDRS), given that this scale is routinely administered and documented in psychiatric health records per clinical practice in our site. With this approach we expect to increase the homogeneity between the retrospective and prospective subcohorts for the primary objective of this study. We have included this clarification in the main text, lines 413-419.

Comment 2: The variables to be assessed in the prospective cohort are clearly outlined. However, it would be helpful to specify which variables are available (or expected to be available) in the retrospective cohort.

Response: We thank Reviewer #1 very much for this important observation. Of note, all the clinical and demographic variables to be assessed in the prospective subcohort have been selected taking into consideration the routine clinical practice of an academic hospital which includes the core variables of this study: the routine use of TDM and pharmacogenetic analyses per clinical judgment, the specified psychometric scales (HDRS, MADRS, HARS, C-SSRS, Y-BOCS and MMSE), and thorough clinical interviews and documentation that allow to retrieve the same core information in the retrospective subcohort. Only minimal and reasonable burden is added in the prospective subcohort relative to routine clinical practice, to obtain relevant information such as the CGI, the BDI, the UKU side effects rating scale, the MARS, the QLDS and the exploration of patients’ expectations about their pharmacological treatment. These latter variables are expected to be available only in the prospective subcohort. We have included this clarification in the main text, lines 308-318.

Comment 3: The analysis plan is rather very brief and would benefit from further elaboration.

Response: We appreciate very much for this crucial correction. Below we provide a response to each specific request regarding the analysis plan.

Specifically:

- What statistical analyses are planned for the retrospective cohort, given the nature and possible limitations of the data available in the medical records?

Response: The analysis approach will be similar for both retrospective and prospective cohorts, based on logistic regressions and correlations (lines 341-348). We anticipate a good data

availability in the medical records for the retrospective cohort, although we cannot fully guarantee the same level of homogeneity for the planned timepoints. To reinforce the homogeneity in data collection, the timepoints of the retrospective cohort will be adjusted as much as possible to the flexible time-window planned for the prospective cohort. We have added on this limitation in the main text, lines 410-415:

“Furthermore, the adherence of the retrospective subcohort to planned timepoints may depend on the availability of data. This may result in lack of homogeneity between retrospective and prospective subcohorts. To address this limitation, separate analysis for both subcohorts will be also conducted. In addition, the timepoints of the retrospective cohort will be adjusted as much as possible to the flexible time-window planned for the prospective cohort.”

- More detail on the logistic regression models would be helpful. Which specific associations are you aiming to explore, and what variables will be included in the models?

Response: We have added more details on the logistic regression models in the revised version of the manuscript. The odds ratio and its 95% confidence interval will be calculated as a measure of association using logistic regression models. These models will be used to assess specifically the associations between blood concentration of sertraline (independent variable) and variables of effectiveness (therapeutic response and remission according to HDRS and MADRS), suicide ideation and/or behavior by means of the C-SSRS, and potential side effects (presence or absence). We have included these details and variables to be included in the models in the main text, lines 344-348:

“The odds ratio and its 95% confidence interval will be calculated as a measure of association using logistic regression models. These models will be used to assess specifically the associations between blood concentration of sertraline (independent variable) and variables of effectiveness (therapeutic response and remission according to HDRS and MADRS), suicide ideation and/or behavior by means of the C-SSRS, and potential side effects (presence or absence).”

- One of the secondary objectives is to assess the association between blood sertraline concentrations and other clinical variables related to effectiveness and side effects. What is the analysis plan for this objective?

Response: We have clarified this point in the revised version of the manuscript. As mentioned above, the association between blood sertraline concentrations and clinical variables related to effectiveness and side effects will be assessed using correlation and logistic regression models (lines 341-352):

“Pearson's correlation coefficient will be used to assess the association between plasma sertraline concentration and the intensity of depressive symptoms. The odds ratio and its 95% confidence interval will be calculated as a measure of association using logistic regression models. These models will be used to assess specifically the associations between blood concentration of sertraline (independent variable) and variables of effectiveness (therapeutic response and remission according to HDRS and MADRS), suicide ideation and/or behavior by means of the C-SSRS, and potential side effects (presence or absence). P values <0.05 will be considered to establish statistical significance. Statistical analyses (including interactions and sensitivity analyses) and will be performed using the available Stata versions in our site, currently up to Stata 19 (StataCorp. 2025. Stata Statistical Software: Release 19. College Station, TX: StataCorp LLC). No imputation for missing data will be applied.”

Reviewer #2:

Thank you for the opportunity to review this important and timely research study. The study presents a comprehensive exploration of bio-psycho social model in the management of patients with depression and offers valuable insight into personalized sertraline treatment approaches, particularly through genetic profiling. This is a relevant and forward-looking topic that could meaningfully contribute to both clinical practice and future research.

Response: We thank Reviewer #2 very much.

Minor Comments

1. Terminology Clarification – "Tolerability" vs. "Drug Safety"

In both the abstract (line 36) and the methods section (line 120), the authors appear to use "tolerability" and "drug safety" interchangeably. However, these terms have distinct meanings in pharmacological research and should be clearly differentiated.

• Drug safety refers to the broader evaluation of all potential adverse effects, including serious adverse events and long-term risks, and entails a comprehensive risk–benefit analysis.

• Drug tolerability, on the other hand, centers on the subjective experience of the patient, especially non-serious side effects that affect adherence or comfort.

I recommend that the authors revise these sections to reflect the appropriate term depending on the specific outcome they are measuring, or explicitly state whether both aspects are being evaluated.

Response: We thank Reviewer #2 very much for this insightful recommendation, which we overlooked and very much agree that should be followed. Given that a comprehensive exploration of side effects will be conducted in every visit of the prospective subcohort, including serious and unexpected adverse effects, we believe that both aspects are being evaluated. The text specifies that “Suspected cases of serious and unexpected adverse reactions to sertraline will be reported through the official channel provided to site’s computerized system and medical records” (lines 358-360). Accordingly, as requested, in the revised version of the manuscript we have now explicitly stated throughout the text that both aspects (tolerability and safety) are being evaluated. This affected the Title (line 3), Abstract (lines 31, 36) and text (lines 102, 107, 121, 188, 360).

2. Feasibility of Assessments Across Age Spectrum

The methods describe multiple assessment tools administered in a single session. Given that the study includes participants across a broad age range including older children and the elderly. it would be helpful to clarify how the burden of lengthy assessments (potentially exceeding two hours) will be managed. Specifically:

• Will assessments be broken into phases or sessions for specific age groups?

• How will participant fatigue or cognitive capacity in extreme age groups be accounted for?

Clarifying the administration logistics would strengthen the feasibility of the methodology.

Response: We thank Reviewer #2 for this observation regarding the logistics, which we have incorporated in the main text. The revised “Description of all processes and interventions from informed consent” now includes the requested clarifications (lines 191-200):

“Steady-state serum samples for the Therapeutic Drug Monitoring of sertraline will be obtained in fasting condition at first hour in the morning. After that, the patients will have the opportunity to have breakfast prior to the clinical assessment. Although only minimal and reasonable burden is added in the prospective subcohort relative to routine clinical practice to obtain relevant information, each clinical assessment may be broken into two sessions in the same day if the patient experiences fatigue. Patients’ fatigue and cognitive capacity during the assessments will be taken into account by the psychiatrists conducting the assessment, and the opportunity to include a resting period at any timepoint of the assessment will be offered per clinical judgment, between two given clinical scales or parts of the psychiatric clinical interview.”

3. Ethical Considerations – Informed Consent and Assent

The manuscript states that informed consent will be obtained from adult participants and the parents/guardians of minors. However, it is important to also address assent, particularly for participants aged 12 and older.

Under Spanish regulations (Law 41/2002), while parental consent is required for minors, children aged 12 and above should also provide their assent. The authors should clarify whether and how assent will be obtained in accordance with these ethical requirements.

Response: We thank Reviewer #2 for the observation of the important differentiation between Informed Consent and Assent under Spanish regulations, which is absolutely right. Indeed, our study includes three different versions of the document including the subject information sheet and informed consent form (all had been already approved by the local Research Ethics Committee): one for adults, a different one for minors, and an additional one for minors aged 12 or above to document assent in that age group. We did not include these specifications in the first version of our manuscript because we considered this whole aspect implicitly covered and understood under the approval by our local, Spanish Research Ethics Committee. However, given the importance of the description of the ethical considerations in a global, international and multicultural scenario, we agree with the reviewer’s request, and the specifications regarding assent have now been included in the revised manuscript (lines 377-383).

4. Data Sharing Policy

The data management plan is well written, but the data sharing statement could be improved. While the authors indicate that data will be available upon request, it is advisable to align with current data sharing policies, which encourage more transparent and accessibl

---

## [Editor Report · Decision Letter 1]

21 Jul 2025

The PREDEP-SERT study protocol: a 6-month follow-up cohort study of predictors of effectiveness, tolerability and safety of sertraline for depression using Therapeutic Drug Monitoring

PONE-D-25-25340R1

Dear Dr. Molero,

We’re pleased to inform you that your manuscript has been judged scientifically suitable for publication and will be formally accepted for publication once it meets all outstanding technical requirements.

Kind regards,

Dickens Akena, Ph.D

Academic Editor

PLOS ONE
---

## [Editor Report · Acceptance letter]

PONE-D-25-25340R1

PLOS ONE

Dear Dr. Molero,

I'm pleased to inform you that your manuscript has been deemed suitable for publication in PLOS ONE. Congratulations! Your manuscript is now being handed over to our production team.

Kind regards,

on behalf of

Dr. Dickens Akena

Academic Editor

PLOS ONE